# Sequence Analysis of the Malaysian Low Pathogenic Avian Influenza Virus Strain H5N2 from Duck

**DOI:** 10.3390/genes14101973

**Published:** 2023-10-22

**Authors:** Fatin Ahmad Rizal, Kok Lian Ho, Abdul Rahman Omar, Wen Siang Tan, Abdul Razak Mariatulqabtiah, Munir Iqbal

**Affiliations:** 1Laboratory of Vaccine and Biomolecules, Institute of Bioscience, Universiti Putra Malaysia, 43400 UPM Serdang, Selangor, Malaysiaaro@upm.edu.my (A.R.O.); 2Department of Pathology, Faculty of Medicine and Health Sciences, Universiti Putra Malaysia, 43400 UPM Serdang, Selangor, Malaysia; klho@upm.edu.my; 3Department of Veterinary Pathology and Microbiology, Faculty of Veterinary Medicine, Universiti Putra Malaysia, 43400 UPM Serdang, Selangor, Malaysia; 4Department of Microbiology, Faculty of Biotechnology and Biomolecular Sciences, Universiti Putra Malaysia, 43400 UPM Serdang, Selangor, Malaysia; wstan@upm.edu.my; 5Department of Cell and Molecular Biology, Faculty of Biotechnology and Biomolecular Sciences, Universiti Putra Malaysia, 43400 UPM Serdang, Selangor, Malaysia; 6Avian Influenza and Newcastle Disease Group, The Pirbright Institute, Woking GU24 0NF, UK

**Keywords:** Avian influenza, virus, hemagglutinin, neuraminidase, matrix gene, H5N2

## Abstract

The avian influenza viruses (AIV) of the H5 subtype have the ability to mutate from low pathogenic (LPAI) to highly pathogenic (HPAI), which can cause high mortality in poultry. Little is known about the pathogenic switching apart from the mutations at the haemagglutinin cleavage site, which significantly contributes to the virus virulence switching phenomenon. Therefore, this study aimed to compare the molecular markers in the *haemagglutinin* (*HA*), *neuraminidase* (*NA*), and *matrix* (*M*) genes of a locally isolated LPAI AIV strain H5N2 from Malaysia with the reference HPAI strains using bioinformatics approaches, emphasising the pathogenic properties of the viral genes. First, the H5N2 strain A/Duck/Malaysia/8443/2004 was propagated in SPF eggs. The viral presence was verified by haemagglutination assay, RT-PCR, and sequencing. Results showed successful amplifications of *HA* (1695 bp), *NA* (1410 bp), and *M* (1019 bp) genes. The genes were sequenced and the deduced amino acid sequences were analysed computationally using MEGA 11 and NetNGlyc software. Analysis of the HA protein showed the absence of the polybasic cleavage motif, but presence of two amino acid residues that are known to affect pathogenicity. There were also two glycosylation sites (glycosites) compared to the reference HPAI viruses, which had three or more at the HA globular head domain. No NA stalk deletion was detected but the haemadsorbing and active centres of the studied NA protein were relatively similar to the reference HPAI H5N2 isolates of duck but not chicken origins. Six NA glycosites were also identified. Finally, we observed a consistent M1 and M2 amino acid sequences between our LPAI isolate with the other HPAI H5N1 or H5N2 reference proteins. These data demonstrate distinct characteristics of the Malaysian LPAI H5N2, compared to HPAI H5N2 or H5N1 from ducks or chickens, potentially aiding the epidemiological research on genetic dynamics of circulating AIV in poultry.

## 1. Introduction

Avian influenza viruses (AIV) are composed of antisense, single stranded, and segmented RNA, and belong to the *Orthomyxoviridae* family. The virus genome is approximately 13.5 kb in length and contains eight segmented genes, i.e., *haemagglutinin* (*HA*), *neuraminidase* (*NA*), *polymerase base subunits one* and *two* (*PB1* and *PB2*), *polymerase acidic* (*PA*), *nucleoprotein* (*NP*), *matrix proteins* (*M*), and *non*-*structural proteins* (*NS*), which encode ten or eleven proteins, depending on the virus strain [1]. The combination of the surface glycoproteins HA and NA determines the subtypes of influenza A viruses, each of which contains 18 (H1–H18) and 11 (N1–N11) antigens [2]. Apart from the HA-NA subtyping, AIV can be classified according to their ability to cause illness and death in chickens, i.e., low pathogenicity AIV (LPAIV) is generally known to cause no apparent or very moderate clinical disease, such as decreased egg production or ruffled feathers, while high pathogenicity AIV (HPAIV) can cause severe and fatal disease by damaging visceral organs and typically results in up to 100% mortality of chickens by day three [3]. All known HPAIV strains are of the H5 or H7 subtype. As evidenced by HPAI outbreaks in the US (1983–1984) and Mexico (1994), LPAI virus strain H5N2 has the propensity to mutate into HPAI viruses. Although the exact mechanisms underlying this pathotype change are somewhat unclear, several lines of evidence suggest that this is due to the recombination that typically occurred between the *HA* gene and *NP* genes, leading to the insertion of additional amino acids at the HA cleavage site. These conformational changes make the site more accessible to ubiquitous proteases such as furin and plasmin, and hence activate the virulence of the virus [4]. Multiple other independent LPAIV to HPAIV conversion events have been documented since 1959, when the first H5 HPAI virus was found in chicken farms in Scotland [5].

Different HPAI H5Nx (x = N1 − N9) strains cause variable disease severity in different bird species. Older HPAI H5 strains showed significant lethality in chickens, turkeys, and quails but not in ducks. However, with the emergence of the A/Goose/Guangdong/1/96 (Gs/Gd) lineage, H5N1 viruses can cause severe disease and mortality in several domestic and wild birds species, including ducks, geese, gulls, terns, shorebirds, and swans [6]. One of the main contributing factors is t differential binding affinity of different AIVs to the host cell surface receptors; the α-2,3- and α-2,6-sialic acid linked galactoses carry preferential binding avidity for avian or other species, respectively, and are widespread in several organs of both chickens and ducks. However, the chicken tracheal and intestinal epithelial cells express a broader range of sialic acid α-2,3-Gal receptors compared to ducks, which suggests that a wider range of avian type AIV can be sustained in chickens when compared to ducks [7]. Furthermore, ducks can initiate a rapid innate immune response due to rapid activation of major pattern recognition receptors, i.e., TLR7, RIG-I, and MDA-5, compared to chickens upon AIV infection [8]. Although chickens express MDA-5 receptors, the absence of RIG-I receptors, which are important in the recognition of viral RNA, mediates antiviral response and is postulated to cause delayed and inadequate inflammatory cytokine responses. Due to their intrinsic endurance against AIV, ducks may enhance the genetic reassortment of AIV strains, thus facilitating the generation of genetically diverse AIVs.

Malaysia’s first reported HPAI H5N1 outbreak in chickens occured in 2004 in Kelantan state. In the same year, LPAI H5N2 viruses were isolated from ducks in the Perak state [9]. Due to the more pressing factor to contain the severe effect of HPAI AIV, molecular, pathological, and vaccine research was focused more towards the circulating HPAIV H5N1 viruses, while the LPAI H5N2 isolates were used as the serological reference to perform the HA inhibition assays or indirect ELISA [10,11,12]. The LPAI H5N2 strain A/duck/Malaysia/8443/04 was developed as an inactivated whole-virus oil emulsion vaccine but was not effective against HPAI H5N1 strain A/chicken/Malaysia/5858/04 virus challenge in chickens [13]. Therefore, this study aimed to analyse the sequence of the Malaysian H5N2 strain in silico, emphasising the pathogenic properties of the viral genes. The data indicate that the Malaysian LPAI H5N2 isolate possesses a minimal threat in terms of pathogenicity due to the lack of HA cleavage, minimal identity to amino acids known to affect virulence, low number of potential glycosylation sites at the HA globular domain, and absence of the NA stalk deletion.

## 2. Materials and Methods

The LPAI H5N2 virus strain A/Malaysia/Duck/8443/04 (kindly provided by the Veterinary Research Institute (VRI) Ipoh, Perak) was propagated inside 11-days-old embryonated specific-pathogen-free (SPF) chicken eggs (Thai SPF Co. Ltd., Nakhon Nayok, Thailand). After 72 h of 37 °C incubation and 60% humidity, the eggs were chilled at 4 °C for 4 h. Then, the allantoic fluid was harvested, centrifuged (1000× *g*, 10 min, 4 °C), and subjected to HA assay. All wells except the first of a microtitre 96-well V-bottomed plate were filled with 50 µL of PBS (Life Technologies, Carlsbad, CA, USA). An aliquot (100 µL) of the virus sample was added to the first well. Then, 50 µL of the virus was diluted two folds across each well. Finally, each well was filled with 50 µL of 0.8% chicken red blood cells. The plate was then incubated for 30 min at room temperature. The plate was tilted 45 degrees to score the results, and the HA activity was verified by the forming of a tear-shaped stream. The virus titre was defined as the lowest virus concentration that caused complete agglutination of RBC and was denoted by the HA unit (HAU).

Viral RNA was extracted from the allantoic fluid using the PureLink Viral RNA/DNA Kit (Invitrogen, Carlsbad, CA, USA). Prior to the extraction, a carrier RNA/lysis buffer mixture provided in the kit was initially prepared. Then, 310 µL of RNAse-free water was added to 310 µg of lyophilised carrier RNA to make a stock of 1 µg/µL. The lysis buffer was added to the specified amount of carrier RNA stock in a fresh sterile tube, and the mixture was utilised within one hour. The subsequent viral extraction protocol was performed according to the manufacturer’s instructions without modifications.

Then, the RNAs were amplified using the SuperScript III one-step reverse transcriptase (RT)-PCR kit (Thermo Fisher Scientific, MA, USA). To begin, 25 µL of 2× reaction master mix, 10 µL of RNA template, 1.6 µM of each forward and reverse primers (Table 1), and 2 µL of platinum Taq High Fidelity Enzyme mix were added to a total reaction of 25 µL. The PCR protocol was 55 °C for 30 min (generation of cDNA), 94 °C for 2 min (pre-denaturation), followed by 40 cycles of 94 °C for 15 s (denaturation), 54 °C, 58 °C, or 60 °C (annealing, for *M*, *NA* and *HA* genes, respectively), and 68 °C for 1 min (extension). Final extension was conducted at 68 °C for 5 min. Positive amplicons were sent for Sanger sequencing (Apical Scientific Sdn Bhd, Selangor, Malaysia).

The full-length sequences of the isolated genes at the nucleotide and amino acid levels were aligned with closely related sequences obtained from the GenBank (NCBI, https://www.ncbi.nlm.nih.gov accessed on 25 September 2021) and Global Initiative for the Sharing All Influenza Data (GISAID, http://www.gisaid.org accessed on 25 September 2021) using MEGA software version 11 [16]. The quantity and distribution of N-glycosylation sites (glycosites) in HA and NA proteins were predicted using NetNGlyc software (https://services.healthtech.dtu.dk/service.php?NetNGlyc-1.0 accessed on 10 October 2022).

## 3. Results

### 3.1. Virus Isolation and Gene Amplications

The harvested virus stock showed a titration of 256 HA unit per 50 µL, which indicated the successful propagation of duck-originated AIV strain in the chicken embryos. Upon first round of RT-PCR on the extracted genomic viral RNA, only partial fragments of the *HA*, *NA*, and *M* genes were generated, i.e., 475 bp, 395 bp, and 726 bp, respectively. Therefore, a primer walking strategy using in-house oligoes was performed to generate the genes’ full-length sequences (Figure 1). The amplicons corresponded to the expected sizes for *HA* (1695 bp), *NA* (1410 bp), and *M* (1019 bp). These genes were deposited into the NCBI database with accession numbers ON714528 (*HA*), ON729981 (*NA*), and ON714539 (*M*).

### 3.2. Nucleotide Sequence Analysis

The full-length *HA*, *NA*, and *M* genes were subjected to BLAST to observe sequence similarity from the NCBI database (https://blast.ncbi.nlm.nih.gov/Blast.cgi accessed on 15 March 2022). The sequence was 99% matched to accession numbers DQ122147, DQ104702, and EU271940 for *HA*, *NA*, and *M* genes, respectively, from strain A/duck/Malaysia/F189/07/04(H5N2), which were deposited by the Agri-Food and Veterinary Authority of Singapore (AVA). Another strain with 98–99% sequence identity with those genes was the A/duck/Malaysia/F118/08/04(H5N2), which was isolated as part of Singapore’s routine surveillance for AIVs in avian species imported from Malaysia [17]. Other than these, the HA and NA sequences showed lower than 94% and 96% sequence identities, respectively, with other AIV strains such as the Mongolian AIV strain (A/duck/Mongolia/54/2001(H5N2). However, the *M* gene showed a 97% identity to a diverse HA subtypes, including H1, H3, H4, H6, H7, and H10, which suggests that the *M* gene is highly conserved across subtypes [18,19].

### 3.3. Amino acid Sequence Analysis

#### 3.3.1. HA Protein

The pathogenicity of AIV can be indicated by the presence of a specific polybasic sequence site in which the ubiquitous subtilisin-like endoproteases, such as furin, will cleave the full length *HA* gene into HA1 and HA2 fragments to activate the severity of infections [20]. Table 2 shows a comparison of the cleavage motif of the studied *HA* gene compared with other HPAI and LPAI H5 reference genes. We observed that the studied HA cleavage motif was RETR, a motif that lacks the multibasic amino acid sequence and is consistent with other LPAI H5 subtypes, thus confirming the classification of our H5N2 virus sample as LPAI.

Amino acid residues of HA at positions 113, 124, 142, 154, 228, and 233 (numbering according to the first methionine residue) are known to affect the pathogenicity of HPAI viruses in chickens [21]. We performed an amino acid analysis of our LPAI HA based on this and observed an identical aspartic acid (D) at position 113 to all HPAI H5 subtype reference strains and an identical asparagine (N) at position 154 to the A/spotbill duck/Xuyi/18/2005(H5N2) strain (Table 3). Therefore, we suggest that the tendency of the studied virus to pose a threat as HPAI is low due to the minimal residue similarity. Furthermore, the identified amino acids are still relying on the rest of the HA amino acid and the balance of NA sequence contents to improve pathogenicity [22].

The N-linked glycosylation pattern of the *HA* gene was examined in which the subtypes of H5 viruses were subjected to N-glycosylation prediction. The results for the A/duck/Malaysia/8443/2004(H5N2) strain from this study showed 7 N-glycosylation locations, namely N26, N27, N39, N181, N302, N496, and N555. There are two glycosites on the globular domain and five glycosites on the stalk domain (Table 4). In the 1983 Pennsylvania outbreak-isolated H5N2 strain Chicken/Pennsylvania/1/83, the presence of site-specific glycosylation around the HA cleavage site and the effect of its internal genes is linked to virus virulence [23,24]. Meanwhile, the reference HPAI viruses showed additional glycosites at the globular head domain, which may associated with increased pathogenesis [25].

#### 3.3.2. NA Protein

Neuraminidase activity can be influenced by the stalk length, haemadsorbing site, active centre, and the amount of glycosites. A deletion in the NA stalk was shown to favour the selection of H5 and H7 viruses with a polybasic cleavage site in the HA, thus increasing the pathogenic potentials [26]. In this study, no evidence of deletion in the stalk region of the studied and reference deduced NA amino acids of duck origins was observed as opposed to the HPAI H5N2 viruses from chicken hosts (deletions at positions 63–65) (Table 5). This may be among the contributing factors to the severity of AIV infection in chickens rather than ducks.

Analysis of sialic acid binding pockets within the haemadsorbing sites revealed the amino acid at positions 366–373, 399–404, and 431–433 [27] of the studied isolate were similar to the reference HPAI H5N2 isolate of duck but not chicken origins. Meanwhile, the active centres at positions 140–157 differ in three amino acids only [27,28]. The design of NA inhibitors should consider these conserved region of the NA haemadsorbing and active sites (Table 5).

NA glycosites with the N–X–T/S motif (in which X may be any amino acid except proline) were generated (Table 6). The result showed the NA protein of the studied and A/chicken/Hebei/1102-MA/2010 (H5N2) isolates contained six, while others contained seven to eight predicted glycosites. The H9N2 viruses of Eurasian ancestry share seven conserved potential NA glycosites [29]. It is suggested that the addition or deletion of glycosite increases virulence by modifying sialidase activity and triggering tissue tropism and antigenicity.

#### 3.3.3. M Protein

The AIV matrix protein consists of the matrix 1 and 2 proteins (M1 and M2, respectively). Three amino acid alterations in M1, notably N30D, I43M, and T215A, increase the pathogenicity of A/H5N1 subtype AIVs in mice [30]. I43M increased virulence in chickens and ducks [30]. However, the underlying mechanisms for this increased virulence and pathogenicity remain unclear [31]. Our analysis showed these amino acids were consistent in the studied *M1* gene with other H5N1 or H5N2 reference proteins (Table 7). Furthermore, the L26F, I/V27A/T/S, A30V/T/S, and G34E amino acids, which are known to contribute to adamantine resistance in M2 [32], were also identical across isolates. Adamantanes (amantadine and rimantadine) disrupt the early stages of virus propagation by blocking the M2 ion channel, thus resistance against the drug is a significant concern. A preventive strategy for the antiviral drug is via the use of vaccines. For example, the AIV vaccine based upon virus-like particles displaying the M2 ectodomain protein has been shown to offer a broad spectrum of immunogenicity [33,34].

## 4. Discussion

The natural reservoir of AIV is believed to be waterfowl belonging to the orders Anseriformes (ducks, geese, and swans) and Charadriiformes (gulls, terns, and sandpipers). LPAIV occasionally spreads from wild birds to domestic and aquatic fowl and are most likely transmitted from wild birds to poultry via direct or indirect interaction. Interestingly, HPAI or LPAI-infected wild ducks often do not exhibit clinical symptoms as opposed to poultry, hence the sequence comparison among the avian hosts continues to be relevant. In this study, we investigated the molecular sequence of a LPAI AIV strain H5N2 originated from a duck that was isolated in Malaysia within the same year of the country’s first HPAIV H5N1 outbreak in chicken. For sequence analysis, we excluded strains A/duck/Malaysia/F189/07/04(H5N2) and A/duck/Malaysia/F118/08/04(H5N2), which were deposited by the Agri-Food and Veterinary Authority of Singapore (AVA), since they shared 98–99% sequence identity to our studied *HA*, *HA*, and *M* genes. We referred our LPAI isolate of duck origin to the HPAI H5N2 and H5N1 isolates of duck and chicken origins to compare the pattern of amino acid changes, which would signify the genetic dynamics.

Several studies have reported the number of basic residues in the HA0 cleavage site, the mutation of key residues in the receptor binding domain, and the alteration of antigenic sites or N-glycosylation sites (glycosites) as among the factors that influence HA characteristics. HPAI viruses contain two or more consecutive basic amino acid (R or K), while the LPAI viruses contain a single or non-consecutive basic amino acid next to the cleavage site, which is only recognized by trypsin-like enzymes produced on epithelial cells. Removal of the pentabasic motif RRRKK from a HPAI HA H5 sequence from a Malaysian H5N1 isolate had been performed in AIV vaccine research to potentially reduce the biosafety concern [12].

AIVs acquire the ability to persist in target hosts by striking an active balance between attaching to the host and releasing progeny virions. To escape the host’s humoral and cellular immune systems, the potential glycosites in viral envelope proteins can provide identical glycans to those of the host’s cells to mask the antigenic sites. The influenza HA protein is highly glycosylated and the glycans on the HA may be recognized by cellular lectin receptors, enabling the virus to adhere to the cells and enable internalisation. Consequently, the degree of glycosylation of HA is probably crucial for identifying the virus by cellular lectins. Asparagine residues in the NXS/T motif (X can be any amino acid besides proline) are glycosylated through post-translational modification [35]. Numbers, types and locations of glycans vary from virus to virus, which may influence the lectin receptors’ ability to recognize AIVs. An increase in the number of N-glycosites on the globular head of HA H5 causes highly virulent avian influenza to escape from vaccine-induced immunity [36].

Beside the HA molecules, NA is essential for the reproduction and spread of the influenza virus, and *NA* gene alterations are strongly linked to antiviral medication resistance. *NA* is now well-known in regulating the pathogenicity or infectiousness of AIV in synergy with the HA. The biological function of the NA stalk is less known. Previous research has demonstrated that deletions in the NA stalk area affect the replication efficiency of AIV in vivo, extended its host range, decreased its NA enzymatic activity, and in some instances, improved its pathogenicity [37]. Stalk deleted NAs were reported sporadically in some AIV subtypes, e.g., H5N1, H6N1, H7N1, H7N3 and H9N2 [38]. This deletion is often accompanied by observations of variants on the HA protein, such as the addition of glycosites, presumably to maintain the functional balance between HA and NA, which is necessary for viral infectivity. These HA variations may have further effects on the antigenicity, virulence and pathogenicity of the virus.

## 5. Conclusions

Based on our findings, we concluded that the Malaysian-like LPAI H5N2 isolate poses a minimal threat in terms of pathogenicity due to the lack of HA cleavage, minimal identity to amino acids known to affect virulence, low number of glycosites at the HA globular domain, and absence of NA stalk deletion. However, the influence of lower number of glycosites and similarity of the haemadsorbing sites and active centres (sites) between the studied LPAI with HPAI NA N2 protein needs laboratory evaluation. It is also worth to note that the amino acid markers in the M1 protein known to increase the AIV pathogenicity were consistent in ducks and chickens, suggesting the M protein is highly conserved and can serve as a potential universal vaccine across avian species.

## Figures and Tables

**Figure 1 genes-14-01973-f001:**
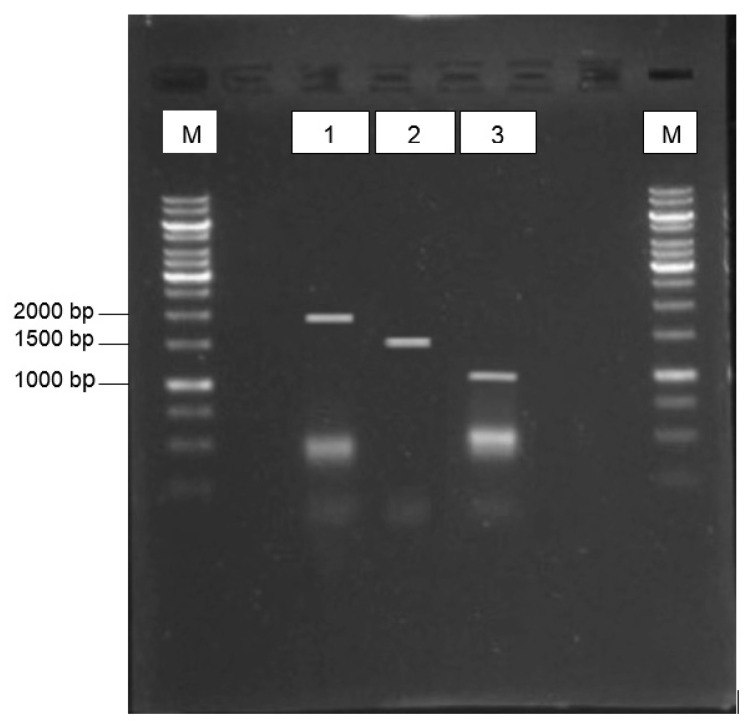
RT-PCR-amplified *HA*, *NA* and *M* genes of the A/Malaysia/Duck/8443/04 H5N2 AIV isolate observed on a 1% (*w*/*v*) agarose gel. Lanes 1, 2, and 3 indicate *HA*, *NA*, and *M* genes, respectively. Lanes M: 1 kb DNA ladder marker (Invitrogen). The amplicons corresponded to the expected sizes for *HA* (1695 bp), *NA* (1410 bp), and *M* (1019 bp).

**Table 1 genes-14-01973-t001:** Primer sequences used for PCR amplification and verification of the *H5N2* genes.

Gene	Direction	Sequence (5′-3′)	Amplicon Size (bp)	Reference
HA	Forward	TATTCGTCTCAGGGAGCAAAAGCAAGGG	1733	[14]
	Reverse	ATATCGTCTCGTATTAGTAGAAA CAAGGGTGTTTT		
NA	Forward	TATTGGTCTCAGGGAGCAAAAGCA GGAGTAGGAGT	1431	[14]
	Reverse	ATATGGTCTCGTATTAGTAGAAAC AAGGAGTTTTTT		
M	Forward	TATTCGTCTCAGGGGCAAAAGCAGGTAG	1027	[14]
	Reverse	ATATCGTCTCGTATTAGTAGAAAC AAGGTAGTTTTT		
N2	Forward Reverse	GCATGGTCCAGTTCAAGTTG CCTTTCCAGTTGTCTCTGCA	362	[15]

**Table 2 genes-14-01973-t002:** Comparison of the HA cleavage site of several HPAI and LPAI H5 viruses.

Strain	Cleavage Site	Molecular Criterion	Reference
(A/duck/Malaysia/8443/2004(H5N2))	PQRE----TRGL	LPAI	This study
(A/duck/Japan/9UO036/2009(H5N2))	PQRE----TRGL	LPAI	JX673924.1
(A/duck/Hokkaido/193/04(H5N3))	PQRE----TRGL	LPAI	AB241625.1
(A/migratoryduck/Jiang Xi/13487/2005(H5N3))	PQRE----TRGL	LPAI	EF597260.1
(A/turkey/Italy/1325/2005(H5N2))	PQRE----TRGL	LPAI	CY022629.1
(A/duck/Malaysia/F118-08-04/2004(H5N2))	PQRE----TRGL	LPAI	DQ104701.1
(A/duck broiler/Malaysia/F189/07/04(H5N2))	PQRE----TRGL	LPAI	DQ122147.1
(A/mallard/Ohio/11OS2229/2011(H5N2))	PQRE----TRGL	LPAI	CY132453.1
(A/duck/New York/483239/2007(H5N2))	PQKE----TKGL	LPAI	GU049935.1
(A/mallard/British Columbia/07826/2005(H5N2))	PQRE----TRGL	LPAI	CY047496.1
(A/duck/Hokkaido/W103/2017(H5N2))	PQRE----TRGL	LPAI	MK592509.1
(A/duck/Hebei/0908/2009(H5N2))	PQIEGR**RRKR**GL	HPAI	JQ041399.1
(A/goose/Guangdong/1/1996(H5N1))	PQRERR**RKKR**GL	HPAI	NC_007362.1
(A/duck/Guangdong/40/2000(H5N1))	PQRERR**RKKR**GL	HPAI	AY585374.1
(A/duck/Guangxi/53/2002(H5N1))	PQRERR**RKKR**GL	HPAI	AY585366.1
(A/duck/Guangzhou/20/2005(H5N1))	PQRERR**RKKR**GL	HPAI	DQ320901.1
(A/duck/Cambodia/D3KP/2006(H5N1))	PQRERR**RKKR**GL	HPAI	HQ200519.1
(A/duck/Thailand/TS04/2006(H5N1))	PQRERR**RKKR**GL	HPAI	JQ794470.1

**Table 3 genes-14-01973-t003:** HA H5 amino acid sequence comparison at positions known to affect pathogenicity.

Virus Strain	Amino Acid Position							Reference
	113	124	142	154	228	233	337–349 (Cleavage Site)	
A/duck/Malaysia/8443/2004(H5N2)	D	T	D	N	S	P	PQRETR---GLF	
(A/duck/Guangzhou/20/2005(H5N1))	D	I	E	L	K	S	PQRERRRKKRGLF	
(A/duck/Thailand/TS04/2006(H5N1))	D	I	E	L	R	S	PQRERRRKKRGLF	
(A/duck/Hebei/0908/2009(H5N2))	D	I	E	L	K	S	PQIEGRRRKRGLF	
A/spotbill duck/Xuyi/18/2005(H5N2)	D	I	E	N	K	S	PQRERRRKKRGLF	

Note: Highlighted rows indicate the HPAI virus strains.

**Table 4 genes-14-01973-t004:** Amino acid comparison of predicted glycosites of H5 HA proteins at the globular and stalk domains.

Virus Strains	Potential N-Gly-Position															
	Globular Domain									Stalk Domain						
	39	88	156	170	179	181	209	252	289	26	27	302	496	500	555	559
A/duck/Malaysia/8443/2004(H5N2)	NVTV	-	-	-	-	NNTN	-	-	-	NNST	NSTE	NSTM	NGTY	-	NGSL	-
A/duck/Hebei/0908/2009(H5N2)	NVTV	NVSE	NPSF	NSTY	NYTN	-	-	-	-	NNST	NSTE	NSSM	-	NGTY	-	NGSL
A/spotbill duck/Xuyi/18/2005(H5N2)	NVTV	-	-	NSTY	-	NNTN	-	-	-	NNST	NSTE	NSSM	-	NGTY	-	NGSL
A/duck/Guangzhou/20/2005(H5N1)	NVTV	-	-	-	-	NNTN	NPTT	-	-	NNST	NSTE	NSSM	-	NGTY	-	NGSL
A/duck/Thailand/TS04/2006(H5N1)	NVTV	-	-	NSTY	-	NNTN	NPTT	-	-	NNST	NSTE	NSSM	-	NGTY	-	NGSL

Note: Highlighted rows indicate the HPAI virus strains.

**Table 5 genes-14-01973-t005:** Amino acid substitutions in the stalk region, haemadsorbing site, and active centre of NA N2 AIV viruses.

Virus	Deletion of the Stalk Region	Haemadsorbing Site			Active Centre
		366–373	399–404	431–433	140–157
(A/Duck/Malaysia/8443/2004(H5N2))	NO	ISKDSRSG	DNSNWS	PQE	LDNKHSNGTIHDRIPHRTL
(A/duck/Hebei/0908/2009(H5N2))	NO	ISKDSRSG	DNSNWS	PQE	LNNKHSNGTIHDRIPNRTL
A/spotbill duck/Xuyi/18/2005(H5N2)	NO	ISKDSRSG	DNNNWS	PQE	LDNKHSNGTIHDRIPHRTL
(A/chicken/Shandong/S3/2014(H5N2))	63–65	IKSDLRSG	DSESWS	PRE	LKNKHSNGTTHDRIPHRTL
(A/chicken/Hebei/1102-MA/2010(H5N2))	63–65	IKSDSRSG	DSDSWS	PQE	LRNKHSNGTTHDRIPHRTL

Note: Highlighted rows indicate the HPAI virus strains.

**Table 6 genes-14-01973-t006:** Sequence of amino acids in glycosites of NA N2 AIV viruses.

Virus Strain	Potential N-Gly-Position															
	61	66	69	70	83	86	143	146	155	197	200	231	234	242	261	402
(A/Duck/Malaysia/8443/2004(H5N2))	NIT	-	NNT	-	-	-	-	NGT	-	-	NAT	-	NGT	-	-	NWS
(A/duck/Hebei/0908/2009(H5N2))	NIT	-	NNT	NTT	-	NWS	-	NGT	NRT	-	NAT	-	NGT	-	-	NWS
A/spotbill duck/Xuyi/18/2005(H5N2)	NIT	-	NNT	NTT	-	-	-	NGT	-	-	NAT	NGT	-	-	-	NWS
A/chicken/Shandong/S3/2014(H5N2))	-	NST	-	-	NWS	-	NGT	-	-	NAT	-	NGT	-	NAS	NVS	-
(A/chicken/Hebei/1102-MA/2010(H5N2))	-	NST	-	-	NWS	-	NGT	-	-	NAT	-	NGT	-	-	NIS	-

Note: Highlighted rows indicate the HPAI virus strains.

**Table 7 genes-14-01973-t007:** Molecular markers of AIV M protein linked to virulence and antiviral resistance.

Virus Strain	Protein and the Specific Amino Acid Markers							
	M1			M2				
	N30D	I43M	T215A	L26F	I/V27A/T/S	A30V/T/S	S31N/G	G34E
	Increased Virulence in Mice	Increased Virulence in Mice, Chickens, and Ducks	Increased Virulence in Mice	Increased Resistance to Amantadine and Rimantadine				
A/duck/Malaysia/8443/2004(H5N2)	D	M	A	L	V	A	S	G
A/duck/Guangzhou/20/2005(H5N1)	D	M	A	-	-	-	-	-
A/duck/Thailand/TS04/2006(H5N1)	D	M	A	L	V	A	N	G
A/duck/Hebei/0908/2009(H5N2)	D	M	A	L	V	A	S	G
A/chicken/Hebei/1102-MA/2010(H5N2))	D	M	A	L	V	A	N	G
A/chicken/Shandong/S3/2014(H5N2)	D	M	A	L	V	A	N	G
A/chicken/Jiangsu/1001/2013(H5N2)	D	M	A	L	V	A	N	G

## Data Availability

The data are available upon request.

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
