# Peer review of "Sequence Analysis of the Malaysian Low Pathogenic Avian Influenza Virus Strain H5N2 from Duck"

_genes, 2023, doi:10.3390/genes14101973_

Round 1

Reviewer 1 Report

The paper titled " Sequence Analysis of the Malaysian Low Pathogenic Avian Influenza Virus Strain H5N2 from Duck " by Fatin Ahmad Rizal et al. concludes that the Malaysian-like LPAI H5N2 isolate poses minimal pathogenicity risks, attributed to factors such as the absence of HA cleavage and limited similarity to virulence-affecting amino acids. The study also notes that the M1 protein, with consistent amino acid markers in ducks and chickens, could serve as a potential universal vaccine for avian species. The work is interesting and suitable for publishing in Genes, but I think some revisions need to be introduced before publication.

Major issues:

Why did the authors choose a flu virus strain from 19 years ago? What is the relevance of this strain to the most recent and prevalent strains?

Minor issues:

1.     Line 116: “The virus titre was defined” should be “The virus titer was defined”.

Minor editing of English language required

Author Response

Dear Reviewer 1, we thank you for your comments which will increase the quality of the present manuscript. Enclosed is our rebuttal. 

Reviewer 2 Report

In the article ” Sequence Analysis of the Malaysian Low Pathogenic Avian Influenza Virus Strain H5N2 from Duck “,  Fatin Ahmad Rizal and al.  present a study, whose aim is to compare the molecular markers in the haemagglutinin (HA), neuraminidase (NA) and matrix (M) genes of a locally isolated LPAI AIV strain H5N2 from Malaysia with the reference HPAI strains.

Avian influenza viruses are still a big concern for animal and human health and all studies concerning this issue are valuable. However, I have a remark. The H5N2 strain A/Duck/Malaysia/8443/2004 was isolated in 2004, which is a long time ago. Therefore, I am not sure about the relevance of the presented information. The remaining remarks concern mainly the methodology.

Lines 54-58, I do not agree with the cited article (3).

Section Materials and Methods. When you describe the methods avoid the word “briefly”. It is clear that the method is described briefly. There is no requirement for a detailed description.

Lines 105-107. Why did you use 11 days old embryonated eggs? Of course, such an age is allowed, but 9 days old are usually used. Is there any particular reason?

Line 110, What do you mean by “standard OIE protocol”? If you would like to cite The OIE Manual of Diagnostic Tests and Vaccines for Terrestrial & Aquatic Animals, you have to write the full title, edition, year, and chapter. And if we stick to that manual, then the HA assay there, is described as follows:

“…Haemagglutination test i) Dispense 0.025 ml of PBS into each well of a plastic V-bottomed microtitre plate. ii) Place 0.025 ml of virus suspension (i.e. infective allantoic fluid) in the first well. For accurate determination of the HA content, this should be done from a close range of an initial series of dilutions, i.e. 1/3, 1/4, 1/5, 1/6, etc. iii) Make twofold dilutions of 0.025 ml volumes of the virus suspension across the plate. iv) Dispense a further 0.025 ml of PBS to each well. v) Dispense 0.025 ml of 1% (v/v) chicken RBCs to each well. vi) Mix by tapping the plate gently and then allow the RBCs to settle for about 40 minutes at room temperature, i.e. about 20°C, or for 60 minutes at 4°C, if ambient temperatures are high, by which time control RBCs should have formed a distinct button….”

In your experiments, you used 50 µl, in OIE Manual described the method with the use of 25 µl of the reagents. Also, the used chicken RBCs are 1%, not 0,8%.

You have to explain your modification, or you have to cite the correct source.

Section Results. Lines 183-190, 265-268, the provided information is more proper for Discussion.

Section Discussion. Lines 297-312, the provided information should be connected with your findings.

Author Response

Dear Reviewer 2, we thank you for your comments which will increase the quality of the present work. Enclosed is our rebuttal. Thank you.
